# The CHARGE syndrome-associated protein FAM172A controls AGO2 nuclear import

Sephora Sallis[1,2], Félix-Antoine Bérubé-Simard[1,*], Benoit Grondin[1,2,*], Elizabeth Leduc[1,2], Fatiha Azouz[1,2], Catherine Bélanger[1], Nicolas Pilon[1,2,3]

**CHARGE syndrome is a neural crest-related disorder mainly caused by mutation of the chromatin remodeler-coding gene *CHD7*. Alternative causes include mutation of other chromatin and/or splicing factors. One of these additional players is the poorly characterized FAM172A, which we previously found in a complex with CHD7 and the small RNA-binding protein AGO2 at the chromatin–spliceosome interface. Focusing on the FAM172A–AGO2 interplay, we now report that FAM172A is a direct binding partner of AGO2 and, as such, one of the long sought-after regulators of AGO2 nuclear import. We show that this FAM172A function mainly relies on its classical bipartite nuclear localization signal and associated canonical importin-$\alpha/\beta$ pathway, being enhanced by CK2-induced phosphorylation and abrogated by a CHARGE syndrome-associated missense mutation. Overall, this study thus strengthens the notion that noncanonical nuclear functions of AGO2 and associated regulatory mechanisms might be clinically relevant.**

## Introduction

CHARGE syndrome is a severe multi-organ developmental disorder mainly affecting derivatives of cranial and cardiac neural crest cells (NCCs) [1, 2]. Most of these anomalies are included in the CHARGE acronym: Coloboma of the eye, Heart defects, Atresia of choanae, Retardation of growth/development, Genital abnormalities, and Ear anomalies. Although heterozygous mutation of *CHD7* (*Chromodomain Helicase DNA Binding Protein-7*) is recognized as the main cause of this syndromic disorder [3], rare variants in many other genes encoding chromatin and/or splicing factors have also been recently associated, including *PUF60, EP300, RERE, KMT2D, KDM6A*, and *FAM172A* [4, 5].

We previously validated the candidacy of *FAM172A* (*Family With Sequence Similarity 172 Member A*) as a CHARGE syndrome-associated gene using a mouse model issued from a forward genetic screen [6]. In this model called *Toupee*, mutagenic transgene sequences are inserted in the last intron of *Fam172a* [4], generating a hypomorphic allele that negatively affects almost every aspect of NCC ontology [4]. Accordingly, homozygous *Toupee* animals (*Fam172a^{Tp/Tp}*) phenocopy both "major" (e.g., coloboma, cleft palate, and vestibular hypoplasia) and "minor" (e.g., retarded growth, cardiac malformations, cranial nerve defects, and genital anomalies) features of the CHARGE syndrome [4, 7]. Moreover, the *Toupee* allele was found to genetically interact with a gene-trap mutant allele of the main CHARGE syndrome-associated gene *Chd7* [4, 8]. Our mechanistic studies further allowed us to propose a model whereby FAM172A—which contains a large ARB2 domain (Argonaute-binding protein 2) split into two halves by a classical bipartite NLS—helps to stabilize the chromatin–spliceosome interface as part of a complex that includes CHD7 and AGO2 [4]. These studies also suggested that perturbation of chromatin-mediated alternative splicing is a general pathological mechanism for CHARGE syndrome, regardless of the involved gene defect [4, 9, 10].

AGO2 (Argonaute-2) is a small RNA-binding protein best known as a major component of the RNA-induced silencing complex, which orchestrates posttranscriptional gene silencing in the cytoplasm [11, 12]. Yet, increasing evidence suggests that AGO2 also fulfills important functions in the nucleus [13], notably for regulating alternative splicing [14, 15, 16, 17, 18, 19, 20]. In agreement with these canonical and noncanonical functions, AGO2 has been reported to shuttle between the cytosol and the nucleus of mammalian cells [17, 21, 22]. The relative subcellular distribution of AGO2 is also known to vary as a function of cell types and culture conditions, being notably generally increased in the nucleus of primary cell cultures as opposed to most immortalized cell lines [23, 24]. Other conditions that increase AGO2 nuclear localization include cellular stress [25, 26], senescence [26, 27], and differentiation [23]. What is less clear is how AGO2 shuttles between both compartments. AGO2 does not contain a NLS or a nuclear export signal (NES). RNA-induced silencing complex-associated TNRC6 proteins

[1]Molecular Genetics of Development Laboratory, Department of Biological Sciences, Université du Québec à Montréal, Montreal, Canada   [2]Centre d'Excellence en Recherche sur les Maladies Orphelines – Fondation Courtois, Université du Québec à Montréal, Montreal, Canada   [3]Department of Pediatrics, Université de Montréal, Montreal, Canada

Correspondence: pilon.nicolas@uqam.ca
Félix-Antoine Bérubé-Simard and Catherine Bélanger's present address is Biopterre, Sainte-Anne-de-la-Pocatière, Canada
*Félix-Antoine Bérubé-Simard and Benoit Grondin contributed equally to this work

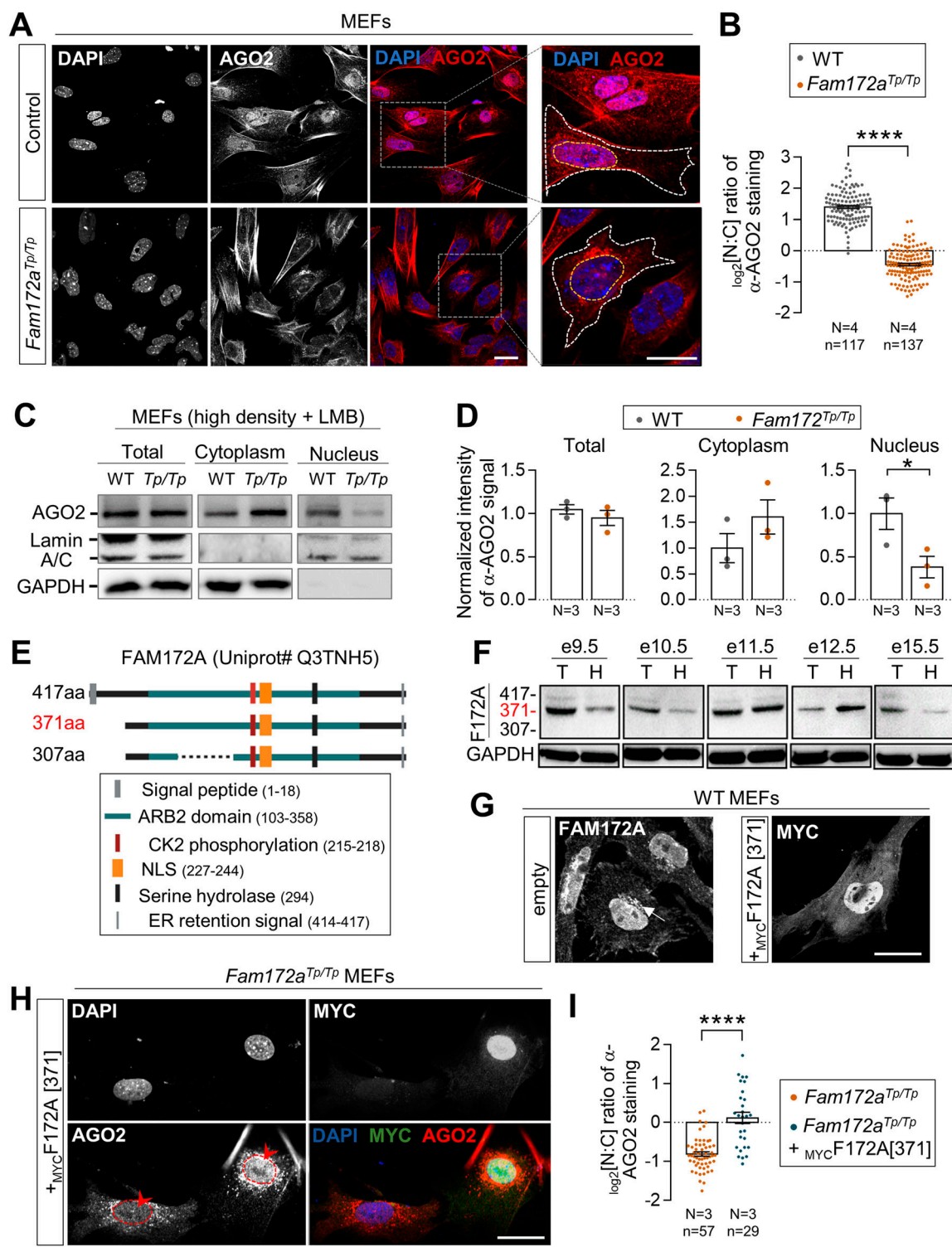

**Figure 1. FAM172A influences AGO2 nuclear localization.**
**(A)** Immunofluorescence analysis of AGO2 distribution in WT and *Fam172a*[Tp/Tp] e10.5 MEFs at moderate density (40,000 cells/cm²), with nuclei stained using DAPI. **(A, B)** Quantification of relative fluorescence intensity of anti-AGO2 staining in the nucleus and cytoplasm (N:C ratio, expressed in log 2 scale) using images such as those displayed in (A). Yellow and white dashed lines in zoomed-in views in (A) delineate measured areas for the nucleus and cytoplasm, respectively. **(C)** Western blot analysis of AGO2 protein levels in cytoplasmic and nuclear fractions of WT and *Fam172a*[Tp/Tp] e10.5 MEFs grown at high density (200,000 cells/cm²) and exposed to leptomycin B showing a specific decrease in the nucleus of *Fam172a*[Tp/Tp] cells (N = 3). GAPDH and Lamin A/C are used as loading control for cytoplasmic and nuclear fractions, respectively. **(C, D)** Quantitative analysis of relative anti-AGO2 signals in mutant versus WT cells (with average in WT set at 1) after normalization with relevant loading control (GAPDH for total and cytoplasmic fractions; Lamin A/C for nuclear fraction), as determined via densitometry (ImageJ) using images such as those displayed in (C). **(E)** The diagram of mouse FAM172A isoforms

(trinucleotide repeat containing 6) contain both an NLS and an NES, thereby making them good candidates as intermediary factors. However, such a role appears limited to AGO2 export only, also involving the major nuclear export factor CRM1 (chromosomal maintenance 1) (28, 29). As of today, the molecular mechanism of AGO2 nuclear import thus remains largely unknown. Although initial studies reported a role for importin-8 (30), follow-up work revealed that many importins are in fact redundantly involved (29) and no NLS-containing protein is currently known to act as intermediary cargo for these importins.

Here, we report that the NLS-containing FAM172A directly interacts with AGO2 and thereby regulates AGO2 nuclear import at least in part via the canonical importin-$\alpha/\beta$ route, a process that we further found to be influenced by the status of CK2-induced phosphorylation of FAM172A.

## Results

### FAM172A influences AGO2 nuclear localization

To determine if FAM172A plays a role in the nucleocytoplasmic shuttling of AGO2, we first compared the subcellular distribution of AGO2 in MEFs freshly derived from $Fam172a^{Tp/Tp}$ and WT e10.5 embryo heads—a cellular model that previously proved useful for highlighting the nuclear colocation of endogenous FAM172A and AGO2 proteins via immunofluorescence (4). As noted in this prior work, AGO2 is enriched in the nucleus of WT MEFs (Fig 1A and B), this enrichment being however lower when MEFs are plated at high density (Fig S1A and B). Under both growth conditions (moderate and high densities), we found that AGO2 is depleted from the nucleus of $Fam172a^{Tp/Tp}$ MEFs, thereby becoming instead markedly enriched in the cytoplasm (Figs 1A and B and S1A and B). A similar outcome was not seen for AGO1, which remains enriched in the nucleus of MEFs despite the loss of FAM172A (Fig S1E and F). Using Western blot and fractionation assays in MEFs grown at high density, we further confirmed that the loss of FAM172A specifically affects AGO2 nuclear localization and not overall protein levels (Figs 1C and D and S1C and D), an outcome that is best evidenced when limiting AGO2 nuclear export using the CRM1 inhibitor leptomycin B (LMB) (31) (Fig 1C and D). Other immunofluorescence data revealed that the converse is not true: the predominantly nuclear localization of FAM172A is not altered by the presence of high levels of transfected $_{FLAG}$AGO2 in the cytoplasm of control MEFs (Fig S1G).

To strengthen our observations, we next sought to rescue AGO2 nuclear depletion in $Fam172a^{Tp/Tp}$ MEFs via transfection of a FAM172A-expressing vector. To this end, we first reviewed the different isoforms of FAM172A, which have been updated since our previous work (4). The Ensembl genome browser now predicts three

major isoforms in mice (Fig 1E), all three being presumably detectable with the commercially available polyclonal antibody used in our laboratory (ab121364; Abcam). Using this antibody to probe embryo extracts at different developmental stages (e9.5 to e15.5) by Western blot, we discovered that the 371aa-long isoform is the most predominant, whereas the 417aa- and 307aa-long isoforms are weakly expressed and undetectable, respectively (Fig 1F). The same pattern was observed in several murine cell lines such as Neuro2a (N2a) neuroblasts, NIH 3T3 fibroblasts, and R1 embryonic stem cells (Fig S1H). Accordingly, we found that a MYC-tagged version of the 371aa isoform can largely recapitulate the subcellular distribution of endogenous FAM172A in MEFs (Fig 1G), except for the ER that is most likely targeted by the signal peptide-containing 417aa-long isoform (4). It should be noted that immunofluorescence was our best option for these analyses, because of very low transfection efficiency in MEFs—which notably prevented us from using cell fractionation and Western blot that require large amounts of cells. Most importantly, when this MYC-tagged version of the 371aa-long isoform (hereinafter referred to as $_{MYC}$FAM172A) was transiently transfected in $Fam172a^{Tp/Tp}$ MEFs, it could reestablish the predominantly nuclear localization of AGO2 (Fig 1H and I)—albeit not up to levels seen in WT MEFs (Fig 1B). Hence, we conclude that FAM172A is actively involved in the regulation of AGO2 nuclear localization.

### FAM172A directly interacts with AGO2

Co-immunoprecipitation (co-IP) assays in multiple cell types/ tissues previously showed that FAM172A and AGO2 from both endogenous and exogenous sources can be found in a same complex, but not when using a version of murine FAM172A mimicking the CHARGE syndrome-associated human variant pE228Q (pE229Q in mice) (4). To formally determine if the FAM172A–AGO2 interaction is direct, we performed in vitro co-IP assays using purified $_{MBP}$FAM172A and $_{His}$AGO2 recombinant proteins. Analysis of both WT and E229Q-mutated $_{MBP}$FAM172A in this context revealed that the FAM172A–AGO2 interaction is direct and further confirmed the prominent role played by the NLS-located E229 residue of FAM172A, although its mutation did not completely abrogate the interaction with AGO2 in these fairly permissive conditions (Fig 2A). To narrow down the interaction interface in AGO2, we also performed co-IP assays using whole lysates of N2a cells transfected with $_{MYC}$FAM172A and deletion constructs of $_{FLAG}$AGO2. This analysis identified the PAZ domain-containing N-terminal half of AGO2 (amino acids 1–480) as being essential for mediating the interaction with FAM172A, with very little contribution by the PIWI-containing C-terminal half (amino acids 478–860) (Fig 2B and C). Consistent with the requirement of this interaction for AGO2 nuclear localization, double-immunofluorescence staining of $_{MYC}$FAM172A-transfected N2a cells

based on Ensembl/Uniprot databases. **(F)** Western blot analysis of FAM172A isoforms in trunk (T) or head (H) extracts from embryos at the indicated developmental stages, using GAPDH as loading control (N = 3). **(G)** Immunofluorescence staining of endogenous FAM172A and transfected MYC-tagged version of the 371aa-long isoform ($_{MYC}$F172A[371]) in e10.5 MEFs, with nuclei stained using DAPI (N = 3). The arrow in the left panel points the endoplasmic reticulum. **(H)** Immunofluorescence analysis of AGO2 distribution in $Fam172a^{Tp/Tp}$ e10.5 MEFs transfected or not with $_{MYC}$F172A[371]. Red arrowheads compare fluorescence intensity in nuclei of transfected and non-transfected cells. **(H, I)** Quantification of relative fluorescence intensity of anti-AGO2 staining in the nucleus and cytoplasm (N:C ratio, expressed in log 2 scale) using images such as those displayed in (H). Scale bar, 20 $\mu$m. N = number of biological replicates, n = number of cells. *$P \leq 0.05$ and ****$P \leq 0.0001$; t-test. (Further data can be found in Fig S1).

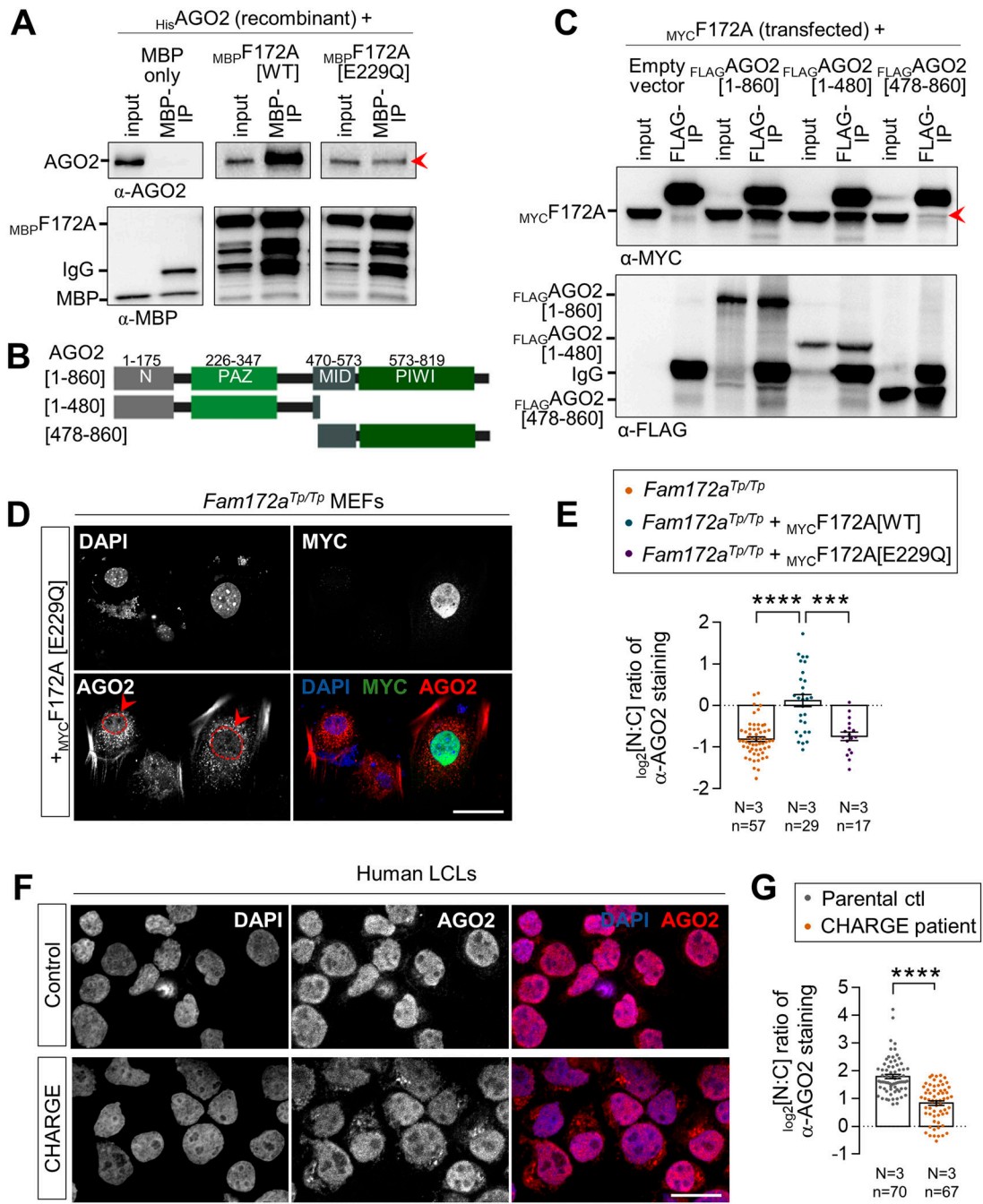

**Figure 2. FAM172A directly interacts with AGO2.**

**(A)** In vitro co-IP of recombinant MBPFAM172A (WT *versus* E229Q versions) and HisAGO2 using MBP as bait (N = 3). MBP alone was used as negative control. The red arrowhead points to reduced amount of co-immunoprecipitated HisAGO2 when using the E229 version of FAM172A. **(B, C)** Co-IP of transfected MYCFAM172A and FLAGAGO2 (full-length *versus* indicated N-term and C-term truncations; see panel (B)) in N2a cells using FLAG as bait (N = 3). The red arrowhead in panel C points to reduced amount of co-immunoprecipitated MYCFAM172A when using the C-term half of AGO2. **(D)** Immunofluorescence analysis of AGO2 distribution in *Fam172a^{Tp/Tp}* e10.5 MEFs transfected or not with E229Q-mutated MYCFAM172A, with nuclei stained using DAPI. Red arrowheads compare relative fluorescence intensity in nuclei of transfected and non-transfected cells. **(D, E)** Quantification of relative fluorescence intensity of anti-AGO2 staining in the nucleus and cytoplasm (N:C ratio, expressed in the log 2 scale) using images such as those displayed in (D). Data for *Fam172a^{Tp/Tp}* and *Fam172a^{Tp/Tp}* + MYCF172A[WT] conditions are the same as initially displayed in Fig 1I (both assays were performed at the same time), being duplicated here for comparison purposes only. **(F)** Immunofluorescence analysis of AGO2 distribution in human LCLs derived from an individual with FAM172A[E228Q]-associated CHARGE syndrome and parental control, with the nuclei stained using DAPI (N = 3 technical replicates). **(F, G)** Quantification of relative fluorescence intensity of anti-AGO2 staining in nucleus and cytoplasm (N:C ratio, expressed in the log 2 scale) using images such as those displayed in (F). Scale bar, 20 μm. N = number of biological replicates, n = number of cells. ***P ≤ 0.001 and ****P ≤ 0.0001; *t*-test. (Further data can be found in Fig S2).

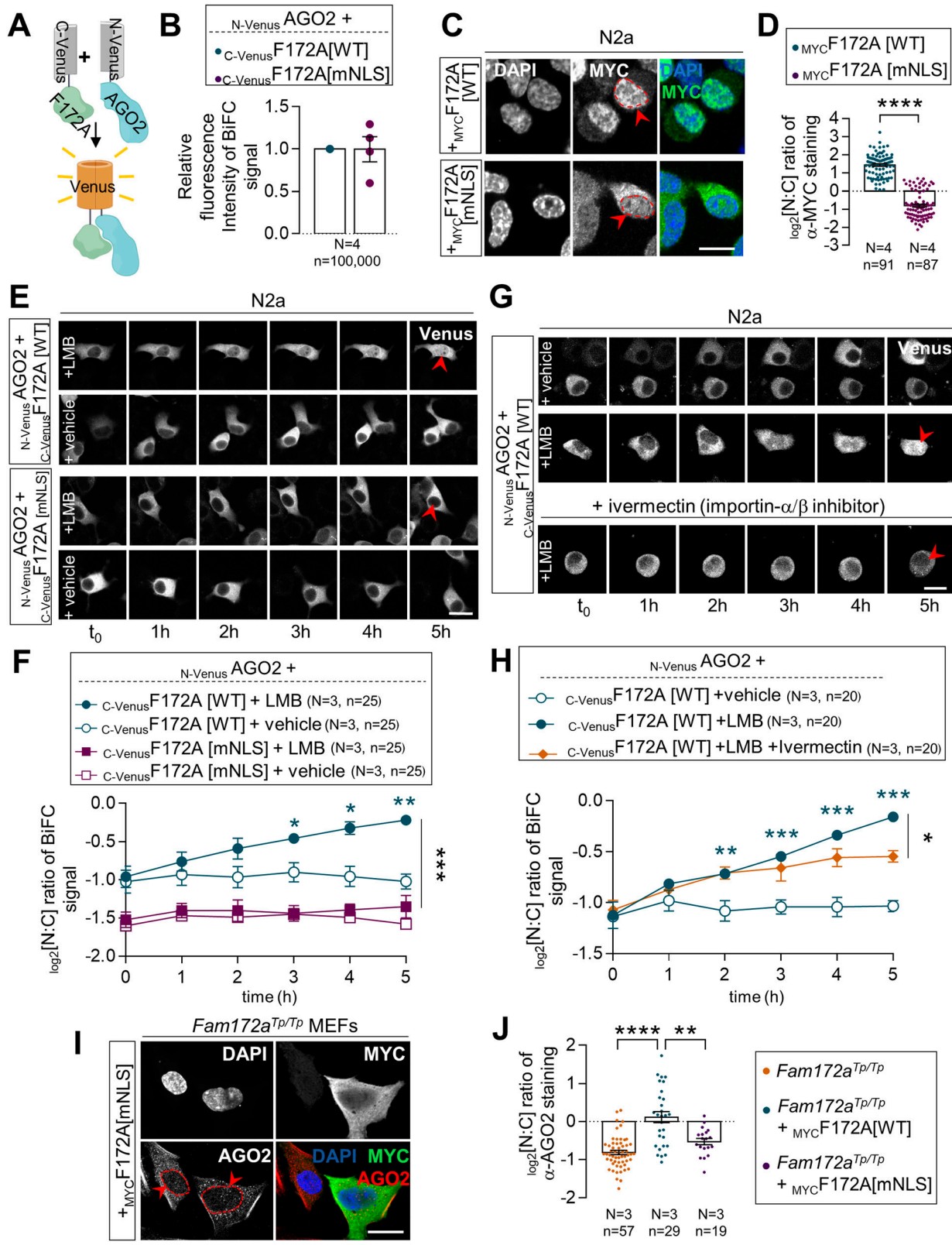

**Figure 3. The NLS of FAM172A is required for AGO2 nuclear import.**
**(A)** Diagram of the BiFC assay based on structural complementation between two non-fluorescent N-terminal and C-terminal halves of a yellow fluorescent protein (Venus) that are respectively fused to AGO2 ($_{N-Venus}$AGO2) and FAM172A ($_{C-Venus}$F172A). **(B)** Comparison of BiFC fluorescence intensity between $_{N-Venus}$AGO2–$_{C-Venus}$F172A[WT] and $_{N-Venus}$AGO2–$_{C-Venus}$F172A[mNLS]. Mean fluorescence intensity was determined using flow cytometry 24 h after transfection in N2a cells. **(C)** Immunofluorescence

showed that nuclear signal intensity for the co-transfected PAZ domain-containing N-terminal half is similar to full-length $_{FLAG}$AGO2, being in both cases higher than for the PIWI-containing C-terminal half (Fig S2A and B). However, both truncated versions of $_{FLAG}$AGO2 gave widely spread out data values in this assay and hence were not found to be statistically different from each other (Fig S2B). This suggests that the minimal contribution of the PIWI-containing half of AGO2 to FAM172A binding might nonetheless be important for efficient nuclear import.

To further evaluate the functional relevance of the direct FAM172A–AGO2 interaction for AGO2 nuclear localization, we then reiterated the rescue experiment in *Fam172a$^{Tp/Tp}$* MEFs. Strikingly, in contrast to WT $_{MYC}$FAM172A (Fig 1H and I), we found that E229Q-mutated $_{MYC}$FAM172A is unable to correct the abnormally enriched cytoplasmic distribution of AGO2 in these cells (Fig 2D and E). Of note, this lack of effect is not because of a reduced ability of E229Q-mutated $_{MYC}$FAM172A to reach the nucleus (Fig 2D). Quantitative analyses in N2a cells even suggest that the E229Q mutation increases $_{MYC}$FAM172A nuclear localization (Fig S2C and D). Collectively, these data thus clearly show that AGO2 nuclear localization relies on a direct interaction between FAM172A and AGO2, which is chiefly mediated by the E229 residue of FAM172A and the PAZ domain-containing half of AGO2. Importantly, this mechanism is most likely at play in human cells as well, as evidenced by the reduced nucleus:cytoplasm ratio of AGO2 in a lymphoblastoid cell line derived from a CHARGE syndrome child bearing the FAM172A variant E228Q (Fig 2F and G).

## The classical bipartite NLS of FAM172A is required for AGO2 nuclear import

To determine if FAM172A either plays an active role in AGO2 nuclear import or only prevents nuclear AGO2 to leave the nucleus, we turned to bimolecular fluorescence complementation (BiFC) assays (32). To this end, we generated a series of fusion proteins where the N-terminal half of Venus (a brighter variant of YFP; yellow fluorescent protein) is located on either end of AGO2 and the C-terminal half of Venus is located on either end of FAM172A (Fig 3A). Upon transfection in N2a cells, three out of the four tested combinations allowed to reconstitute a fluorescent Venus protein (Fig S3A). The combination where both AGO2 and FAM172A have their respective Venus half at their N-terminal end ($_{N-Venus}$AGO2 and $_{C-Venus}$FAM172A) gave a more robust fluorescent signal and was thus

selected for further studies. Using flow cytometry, we confirmed the specificity of this BiFC signal using an E229Q-mutated version of $_{C-Venus}$FAM172A as negative control (Fig S3B) and by competition with an excess of $_{FLAG}$AGO2 (Fig S3C). Using immunofluorescence, we also verified that the nonfluorescent fragments fused to AGO2 and FAM172A do not alter their respective subcellular distribution in N2a cells (Fig S3D). Initial observations of the subcellular distribution of the BiFC signal generated by the $_{N-Venus}$AGO2–$_{C-Venus}$FAM172A complex in living N2a and NIH 3T3 cells showed an enrichment in the cytoplasm after 48 h of culture (Fig S3A and E). This outcome was somehow surprising considering our previous co-IP experiments using N2a cell fractions (cytoplasm versus nucleus) that detected an AGO2–FAM172A complex in the nuclear fraction only (4). We interpret this apparent discrepancy to the methods used, with BiFC being able to capture and lock transient interactions that cannot be detected using co-IP (32).

The cytoplasmic enrichment of the BiFC signal supported the hypothesis that AGO2 and FAM172A initiate their interaction in the cytosol and then together enter the nucleus using the classical bipartite NLS of FAM172A (K/R-K/R-X$_{10–12}$-K/R$_{3/5}$, Fig S4A)—thereby also predicting a role for the canonical importin-$\alpha/\beta$ route (33). To test this model, we devised an experiment based on time-lapse imaging and LMB-induced blockage of nuclear export, also comparing WT- and NLS-mutated versions of $_{C-Venus}$FAM172A. In preparation for this experiment, we carefully point-mutated basic residues in each lysine/arginine-rich half of the bipartite NLS sequence of FAM172A (Fig S4A) to impair its nuclear import without affecting its capacity to interact with AGO2 (Fig 3B–D). N2a cells co-transfected with $_{N-Venus}$AGO2 and either WT- or NLS-mutated versions of $_{C-Venus}$FAM172A were then followed under a confocal microscope during 5 h after addition of LMB. Remarkably, LMB addition led to a constant increase of the BiFC signal in the nucleus over time, but only when using WT $_{C-Venus}$FAM172A (Fig 3E and F and Video 1). With NLS-mutated $_{C-Venus}$FAM172A, the BiFC signal already appeared more enriched in the cytoplasm before LMB addition and remained as such after LMB addition (Fig 3E and F and Video 2). To further validate the involvement of the canonical importin-$\alpha/\beta$ route, we reiterated the BiFC assay using $_{N-Venus}$AGO2 and the WT version of $_{C-Venus}$FAM172A in LMB-treated N2a cells, but now, in the additional presence of ivermectin—a small molecule that inhibits importin-$\alpha/\beta$ complexes by specifically targeting importin-$\alpha$ (34). From 3 h posttreatment , we found that ivermectin blunted the otherwise steady time-dependent increase of BiFC signal in the

analysis of MYC-tagged WT and NLS-mutated FAM172A proteins in transfected N2a cells, with nuclei stained using DAPI. Red arrowheads compare relative fluorescence intensity in the nucleus for WT- and NLS-mutated FAM172A. **(C, D)** Quantification of relative fluorescence intensity of anti-MYC staining in the nucleus and cytoplasm (N:C ratio, expressed in the log 2 scale) using images such as those displayed in (C). **(E)** 5-h-long time-lapse recordings of BiFC signal generated using $_{N-Venus}$AGO2 and $_{C-Venus}$F172A (comparing WT and mNLS versions) in N2a cells treated with leptomycin B or vehicle (ethanol) only. Red arrowheads compare relative fluorescence intensity in the nucleus. **(E, F)** Quantification of relative BiFC signal intensity in the nucleus and cytoplasm (N:C ratio, expressed in log 2 scale) using images such as those displayed in (E). **(G)** Five-hour-long time-lapse recordings of BiFC signal generated using $_{N-Venus}$AGO2 and $_{C-Venus}$F172A in N2a cells treated with leptomycin B with or without ivermectin or vehicle (ethanol) only. Red arrowheads compare relative fluorescence intensity in the nucleus. **(G, H)** Quantification of relative BiFC signal intensity in the nucleus and cytoplasm (N:C ratio, expressed in log 2 scale) using images such as those displayed in (G). **(I)** Immunofluorescence analysis of AGO2 distribution in *Fam172a$^{Tp/Tp}$* e10.5 MEFs transfected or not with $_{MYC}$F172A[mNLS], with the nuclei stained using DAPI. Red arrowheads compare relative fluorescence intensity in nuclei of transfected and non-transfected cells. **(I, J)** Quantification of relative fluorescence intensity of anti-AGO2 staining in the nucleus and cytoplasm (N:C ratio, expressed in log 2 scale) using images such as those displayed in (I). **(E, G)** Data for *Fam172a$^{Tp/Tp}$* and *Fam172a$^{Tp/Tp}$* + $_{MYC}$F172A[WT] conditions are the same as initially displayed in Fig 1I (both assays were performed at the same time), being duplicated here for comparison purposes only. Scale bar, 20 $\mu$m (E, G). N = number of biological replicates, n = number of cells. *$P \leq 0.05$, **$P \leq 0.01$, ***$P \leq 0.001$, and ****$P \leq 0.0001$; *t*-test. (Further data can be found in Fig S3).

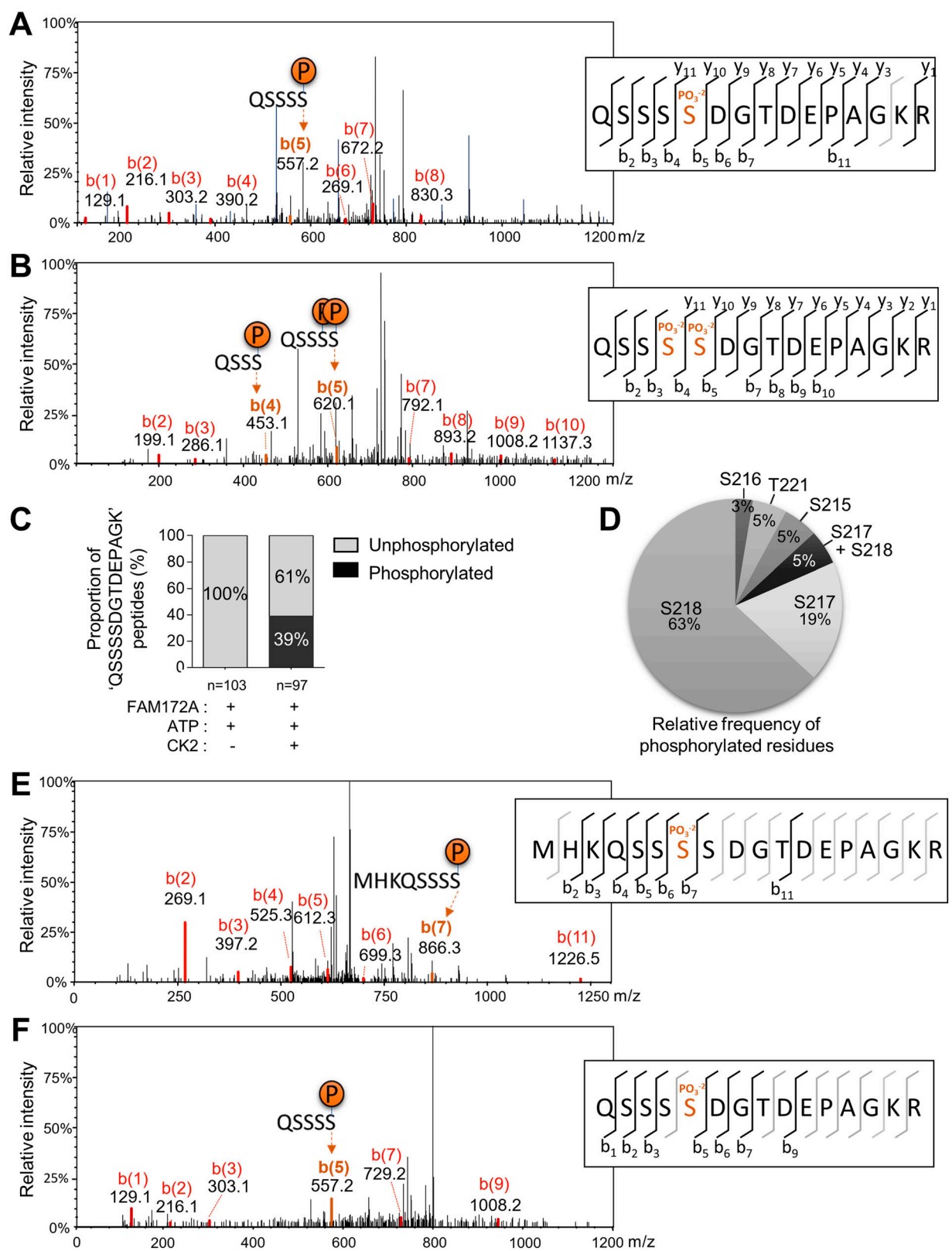

**Figure 4. CK2 phosphorylates FAM172A.**
**(A, B)** Representative CID-MS/MS spectra of most frequently phosphorylated peptides after in vitro co-incubation of recombinant FAM172A with recombinant CK2 and ATP. **(C, D)** Proportion of overall phosphorylation (C) and relative frequency of phosphorylated residues (D) after in vitro CK2-induced phosphorylation. **(E, F)** Representative CID-MS/MS spectra of phosphorylated FAM172A peptides after anti-MYC immunoprecipitation from _MYC_FAM172A-transfected N2a cells.

nucleus of N2a cells treated with LMB only (Fig 3G and H and Video 3). Altogether, these data are consistent with the notion that FAM172A-mediated nuclear import of AGO2 involves an interaction between the classical bipartite NLS of FAM172A and importin-α/β complexes, as further demonstrated by the inability of our NLS-mutated version of $_{MYC}$FAM172A to reestablish AGO2 nuclear enrichment in $Fam172a^{Tp/Tp}$ MEFs (Fig 3I and J).

### The status of FAM172A phosphorylation influences AGO2 nuclear import

The FAM172A interactome in N2a cells is known to include the catalytic subunit alpha of casein kinase II (CK2) (4). Moreover, we noted that FAM172A bears two juxtaposed consensus-like sequences for CK2 phosphorylation (S/T–X–X–D/E) close to its NLS (Fig S4A), and an in vitro assay combined to mass spectrometry analysis confirmed that this motif can indeed be directly phosphorylated by CK2 (Fig 4A–D). Publicly available high-throughput mass spectrometry data (www.phosphosite.org) and our own mass spectrometry analysis of N2 cells (Fig 4E and F) also confirmed that this serine-rich motif (S215-S-S-D-S-S-D in humans; S215-S-S-S-D-G-T-D in mice) can be phosphorylated in a natural cellular context. These data further revealed some variance in the identity of the phosphorylated residue, although S217 and S218 appear preferentially targeted in murine FAM172A—representing 87% (including 5% of double-phosphorylation) and 100% of relevant phosphorylated peptides in vitro (Fig 4A–D) and in N2a cells (Fig 4E and F), respectively.

Given that the proximity of a CK2 phosphorylation site was previously reported to influence NLS recognition by importins (35, 36, 37), we thus wondered if the status of CK2-induced phosphorylation of FAM172A might play a regulatory role in the nuclear translocation of AGO2. To evaluate this possibility, we again took advantage of our BiFC-based live-cell imaging system to now monitor the nuclear import of $_{N-Venus}$AGO2–$_{C-Venus}$FAM172A complexes in LMB-treated N2a cells also exposed to the CK2 inhibitor 4,5,6,7-tetrabromobenzotriazole (38). This inhibitor significantly reduced the nuclear import of the BiFC complex compared with N2a cells treated with LMB only (Fig 5A and B and Video 4). To complement this analysis, we also generated phosphodead (P–) and phosphomimetic (P+) versions of murine FAM172A, in which, either the aspartate residues D220/D223 were replaced by glutamines (39, 40) or the upstream serine/threonine residues were replaced by aspartates (41), respectively (Fig S4A). These new constructs were then tested for their ability to mediate AGO2 nuclear import, once again using the BiFC system in living N2a cells. With the phosphodead $_{C-Venus}$FAM172A, we found that the BiFC signal generated in conjunction with $_{N-Venus}$AGO2 remains enriched in the cytoplasm regardless of the presence of LMB or not (Fig 5C and D and Video 5)—just like when we used the NLS-mutated $_{C-Venus}$FAM172A (Fig 3E and F and Video 2). In stark contrast, the BiFC signal appears greatly enriched in the nucleus when using the phosphomimetic $_{C-Venus}$FAM172A (Fig 5C and D and Video 6), with both basal and LMB-induced levels being greater than for WT $_{C-Venus}$FAM172A (Fig 3E and F and Video 1). A similar outcome was observed when assessing the subcellular distribution of $_{FLAG}$AGO2 in N2a cells co-transfected with $_{MYC}$FAM172A

[P+], which was found to be more efficient than $_{MYC}$FAM172A[WT] at increasing the nuclear localization of $_{FLAG}$AGO2 (Fig S5).

Of note, our BiFC assays further revealed that both phosphodead and phosphomimetic versions of $_{C-Venus}$FAM172A specifically affect the cytoplasmic *versus* nuclear distribution of the $_{N-Venus}$AGO2–$_{C-Venus-}$FAM172A BiFC signal, not overall intensity (Fig 5E). Analysis of the subcellular distribution of phosphodead and phosphomimetic versions of $_{MYC}$FAM172A also suggests that CK2-induced phosphorylation of FAM172A is stimulatory but apparently not essential for its own entry in the nucleus (Fig 5F and G). Nonetheless, this series of experiments strongly suggests that CK2-induced phosphorylation of FAM172A is regulatory important for FAM172A-mediated nuclear import of AGO2.

### AGO2 overexpression can functionally compensate for the loss of FAM172A ex vivo

In a final set of experiments, we tested the functional relevance of FAM172A-mediated nuclear import of AGO2 via rescue experiments in freshly dissociated $Fam172a^{Tp/Tp}$ MEFS, using proliferation rate and alternative splicing of $Cd44$ as cellular and molecular readouts, respectively (Fig S6A and B). To this end, $Fam172a^{Tp/Tp}$ MEFS were transfected with either $_{MYC}$FAM172A (WT, E229Q-mutant or NLS-mutant) or $_{FLAG}$AGO2 (WT or NLS-containing), and then recovered by FACS owing to coexpressed GFP. As previously observed (4), transfection of WT $_{MYC}$FAM172A fully rescued the otherwise decreased proportion of proliferating (i.e., Ki67-positive via immunofluorescence) $Fam172a^{Tp/Tp}$ MEFS compared with WT MEFs (Fig S6A). In contrast, both E229Q-mutated and NLS-mutated versions were totally unable to do so (Fig S6A), consistent with their incapacity to rescue AGO2 nuclear import in these cells (Figs 2D and E and 3G and H). Interestingly, overexpression of full-length WT $_{FLAG}$AGO2 (which increases overall levels in both cytoplasm and nucleus; Fig S1G) partially rescued, whereas another version engineered to contain a strong NLS (three copies of SV40 monopartite NLS; Fig S6C) fully rescued the proliferation rate of $Fam172a^{Tp/Tp}$ MEFS (Fig S6A). Moreover, congruent with their relative ability to interact with FAM172A (Fig 2C) and enter the nucleus (Fig S2A and B), the N-terminal half of $_{FLAG}$AGO2 (PAZ domain-containing; amino acids 1–480) was almost as effective as full-length $_{FLAG}$AGO2 in this assay, whereas the C-terminal half (PIWI domain-containing; amino acids 478–860) had virtually no impact (Fig S6A). Similar results were obtained when assessing alternative splicing of $Cd44$, a known splicing target of AGO2 (14) that we also found to be dysregulated by the loss of either of the CHARGE syndrome-associated proteins FAM172A or CHD7 (4). RT–qPCR analyses revealed higher levels of variable exons 8/9 in $Cd44$ transcripts from $Fam172a^{Tp/Tp}$ MEFS, these abnormal levels being almost fully rescued (i.e., back to normal levels in WT MEFs) upon transfection of WT $_{MYC}$FAM172A or NLS-containing $_{FLAG}$AGO2 (Fig S6B). Altogether, these data are thus in agreement with a general model suggesting that proper FAM172A-mediated nuclear import of AGO2 might be necessary to avoid developing CHARGE syndrome-associated cellular and molecular anomalies.

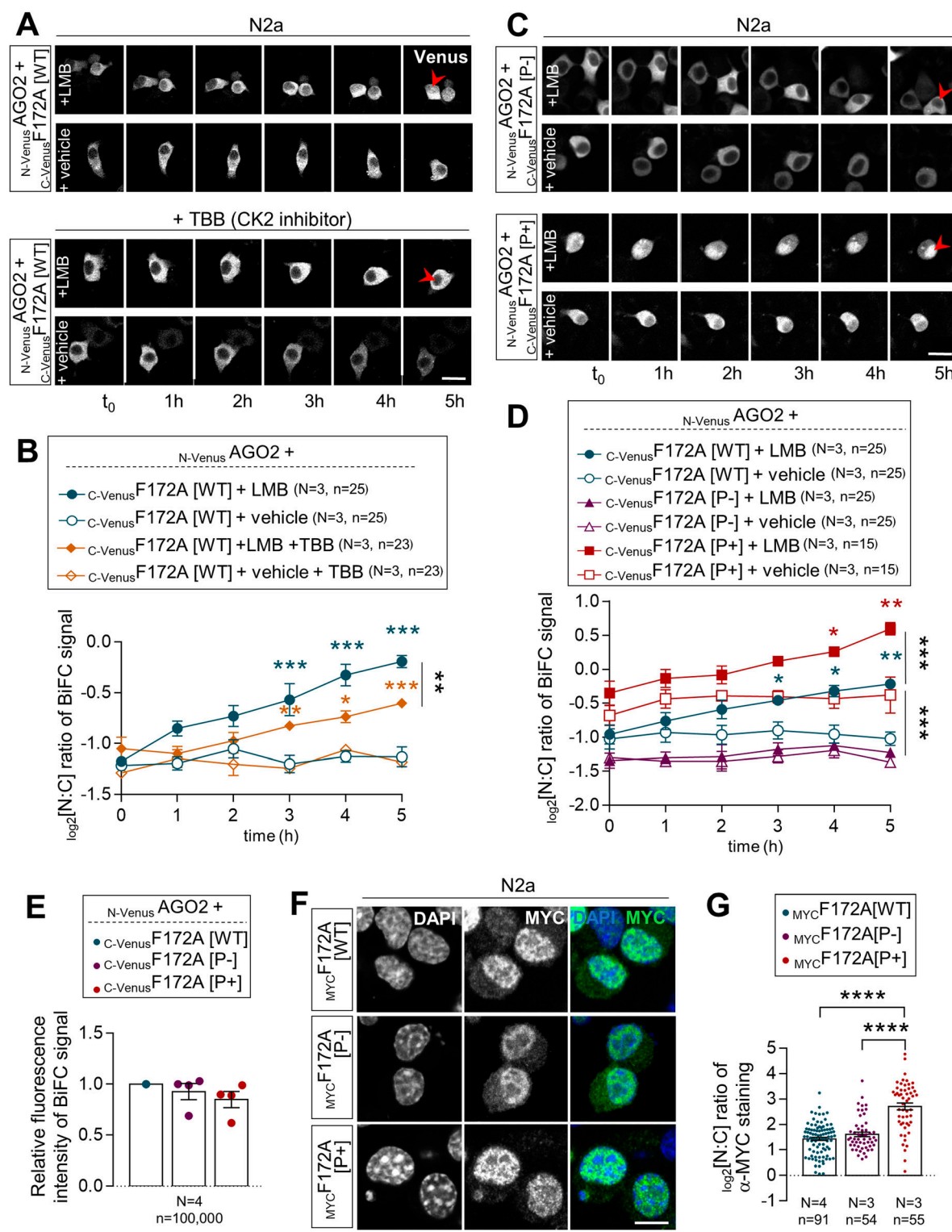

**Figure 5. AGO2 nuclear import depends on the status of FAM172A phosphorylation.**
**(A)** 5-h-long time-lapse recordings of BiFC signal generated using $_{N-Venus}$AGO2 and $_{C-Venus}$F172A in N2a cells treated with leptomycin B or vehicle (ethanol) only, with or without 4,5,6,7-tetrabromobenzotriazole. Red arrowheads compare relative fluorescence intensity in the nucleus. **(A, B)** Quantification of relative BiFC signal intensity in the nucleus and cytoplasm (N:C ratio, expressed in log 2 scale) using images such as those displayed in (A). **(C)** 5-h-long time-lapse recordings of the BiFC signal generated using $_{N-Venus}$AGO2 and $_{C-Venus}$F172A (comparing phosphodead [P−] and phosphomimetic [P+] versions) in N2a cells treated with leptomycin B or vehicle (ethanol) only. Red arrowheads compare relative fluorescence intensity in the nucleus. **(C, D)** Quantification of relative BiFC signal intensity in the nucleus and cytoplasm (N:C ratio, expressed in log 2 scale) using images such as those displayed in (C). Data for $_{C-Venus}$F172A[WT] are the same as initially displayed in Fig 3F (both assays were performed at

# Discussion

To follow up on our prior work and learn more about the poorly characterized CHARGE syndrome-associated protein FAM172A, we here focused on its interaction with the small RNA-binding protein AGO2—which is known to play key gene regulatory roles in the nucleus beyond its canonical function in posttranscriptional gene silencing in the cytoplasm (13). Previously published models predicted a central role for AGO2 at the chromatin–spliceosome interface (14), together with FAM172A that appears required for stabilizing protein–protein interactions (4, 10). Our new data generated using various complementary approaches in murine and patient-derived cells now suggest that FAM172A is in addition a regulator of AGO2 nuclear import, involving both the NLS and CK2 phosphorylation motif of FAM172A.

Although AGO2 has long been known to shuttle between the cytosol and nucleus of mammalian cells (17, 21, 22, 42), we still know very little about the underlying mechanisms. Previous studies suggested that many different NLS-binding importins are likely mediating AGO2 nuclear entry (29, 30), but no NLS-containing proteins were previously identified as intermediary factors for the NLS-free AGO. Such a role has been tested for members of the well-known AGO2-binding protein family TNRC6, which contain both a NLS and a NES (28), with results being consistent with a role in AGO2 nuclear export but not import (28, 29). Notably, knockdown of importin-$\beta$ was shown to specifically impair the noncanonical importin-$\alpha$-independent nuclear import of TNRC6 proteins without perturbing AGO2 nuclear import (29). In contrast, we now present several lines of evidence demonstrating a key role for FAM172A in this process, with BiFC and rescue data from single cells involving its classical bipartite NLS and associated canonical importin-$\alpha$/$\beta$ route (Fig 3). Yet, we do not think that FAM172A is the only factor involved in AGO2 nuclear import. The fact that there are still appreciable levels of AGO2 in the nucleus of $Fam172a^{Tg/Tg}$ MEFs argues in that favor (Fig 1A–D)—although this might also be partly explained by the residual expression of $Fam172a$ (~15% of WT levels) in these hypomorphic mutant cells (4). The existence of other regulators of AGO2 nuclear import would explain, at least in part, why siRNA-mediated knockdown of importin-$\beta$ did not affect AGO2 nuclear import in prior work (29), whereas our FAM172A-centered studies based on NLS mutagenesis and ivermectin treatment did point to a role for the canonical importin-$\alpha$/$\beta$ route (Fig 3). Another factor potentially contributing to this apparent discrepancy might be that the preferred nuclear import pathway for AGO2 varies as a function of cell types (HEK293 in reference 29 versus MEFs and N2a in the current study).

FAM172A itself might also contribute to the diversity of import routes employed by AGO2 – a possibility that would help to explain the rather slow impact of ivermectin in BiFC assays (Fig 3G and H). In line with this, close examination of the linker region that connects each lysine/arginine-rich half of the classical bipartite NLS of FAM172A revealed the additional presence of a K-V-S-K-E-T-K motif (Fig S4A) that matches the consensus valine/isoleucine-dependent binding sequence (K-V/I-X-K-$X_{1-2}$-K/H/R) for Kap121, the yeast ortholog of importin-5 (33, 43). Based on this observation, we tested and confirmed that endogenous importin-5 could be specifically co-immunoprecipitated with $_{MYC}$FAM172A in N2 cells (Fig S7A and B). Yet, we think that these co-IP data should be interpreted with caution, not only because of the rather permissive buffer conditions used (100 mM NaCl ± acute crosslink with 0.75% formaldehyde), but also because the FAM172A interactome includes dozens of nuclear proteins that collectively employ various classes of NLS (4, 9). This latter point most likely explains the otherwise intriguing finding that FAM172A can co-immunoprecipitate importin-$\alpha$1/$\beta$1 even when using our NLS-mutated version (Fig S7C and D) that markedly perturbs FAM172A-mediated nuclear import of AGO2 in single-cell assays (Fig 3E, F, I, and J). In these co-IP assays, we detected a negative effect of the NLS mutation on importin-$\alpha$1 recruitment only once (Fig S7E).

A better understanding of the molecular underpinnings of FAM172A-mediated nuclear import of AGO2 will definitely require additional studies that are beyond the scope of the current study. An important question to address is how FAM172A can interact with both importins and AGO2 given that the E229 residue important for AGO2 interaction (Fig 2) is located within the N-terminal end of the bipartite NLS of FAM172A (Fig S4A). A closely related question to address is how CK2-mediated phosphorylation of FAM172A can enhance AGO2 nuclear import. The AlphaFold-predicted 3D structure of FAM172A (with its NLS and juxtaposed CK2 motif both protruding away from the globular core; Fig S4B) suggests the possibility that the "CK2-NLS loop" might enable interactions with both AGO2 and importins after phosphorylation-induced conformation changes. Alternatively, other experimental data suggest a potential role for FAM172A homodimerization. Indeed, based on the presence of a conserved subregion of the ARB2 domain (Fig S4B) that was previously shown to mediate homodimerization of the yeast proteins Hda1 (44) and Clr3 (45), we tested and validated the possibility that FAM172A can similarly form homodimers using co-IP (Fig S8A) and BiFC (Fig S8B and C and Video 7) assays. Moreover, we found that ivermectin markedly decreased the nuclear enrichment of BiFC signal from FAM172A–FAM172A complexes (Fig S8B and C and Video 8), with a kinetic profile similar to what was observed for AGO2–FAM172A complexes (Fig 3G and H and Video 3). As the putative FAM172A homodimerization interface is away from the CK2-NLS loop (Fig S4B), this suggests that FAM172A homodimers could use an NLS-containing loop to interact with importins on one end and another NLS-containing loop to interact with AGO2 on the other end.

the same time), being duplicated here for comparison purposes only. **(E)** Comparison of overall BiFC fluorescence intensity between $_{N-Venus}$AGO2–$_{C-Venus}$F172A[WT], $_{N-Venus}$AGO2–$_{C-Venus}$F172A[P-], and $_{N-Venus}$AGO2–$_{C-Venus}$F172A[P+]. Mean fluorescence intensity was determined using flow cytometry 24 h after transfection in N2a cells. **(F)** Immunofluorescence analysis of the distribution of MYC-tagged WT, phosphodead (P–), and phosphomimetic (P+) versions of murine FAM172A protein in transfected N2a cells, with nuclei stained using DAPI. **(F, G)** Quantification of relative fluorescence intensity of anti-MYC staining in the nucleus and cytoplasm (N:C ratio, expressed in the log 2 scale) using images such as those displayed in (F). Data for the $_{MYC}$F172A[WT] condition are the same as initially displayed in Fig 3D (both assays were performed at the same time), being duplicated here for comparison purposes only. **(A, C, F)** Scale bar, 20 $\mu$m (A, C) and 10 $\mu$m (F). N = number of biological replicates, n = number of cells. *$P \leq 0.05$, **$P \leq 0.01$, ***$P \leq 0.001$, and ****$P \leq 0.0001$; $t$-test. (Further data can be found in Fig S5).

Clearly, testing both of the above potential mechanisms will represent exciting lines of future research.

Another intriguing aspect of the current study is the fact that the loss of FAM172A in MEFs decreases nuclear levels of AGO2 (Figs 1A–D and S1A–D) but not those of the closely related AGO1 (Fig S1E and F). These observations are consistent with our previous co-IP experiments where we failed to detect an interaction between transfected $_{MYC}$FAM172A and endogenous AGO1 in COS7 cells, despite having used the exact same conditions that worked for endogenous AGO2 (4). However, new data show that FAM172A can interact with AGO1 under certain conditions, like when using recombinant proteins in vitro (Fig S9A) or upon overexpression in N2a cells (Fig S9B). A plausible explanation for all these observations might be that FAM172A simply has a greater affinity for AGO2 than for AGO1, as suggested by the decreased ability of $_{FLAG}$AGO1 to co-immunoprecipitate $_{MYC}$FAM172A in transfected N2A cells compared with $_{FLAG}$AGO2 (Fig S9B).

Interestingly, our data suggesting a regulatory role for CK2 in FAM172A-mediated AGO2 nuclear import (Fig 5) provide a potential mechanistic link to explain why and how different cellular stresses can increase AGO2 translocation in the nucleus (25, 26). Indeed, CK2 activity is known to be enhanced by multiple cellular stressors (46, 47, 48). Moreover, it is noteworthy that about half of the proteins (43%; 29/67 proteins) composing the stress-induced response complex associated with AGO2 in HEK293 cells (25) are also included in the published FAM172A interactome from N2a (4) and/or PGR9E11 cells (9). One protein identified as being of special significance in the stress-induced response complex is nucleolin (25), which we further validated as an interactor for both FAM172A and AGO2 in N2a cells using co-IP (Fig S9C). All these observations further make sense when considering the particular vulnerability of NCCs to cellular stress in general, along with the extensive overlap between neurocristopathies and spliceosomopathies (2, 10, 49). Stress-induced and FAM172A-mediated nuclear import of AGO2 might be a way for this unique, multipotent, and highly migratory cell population to defend against cellular stressors.

# Materials and Methods

## Animals

All experimental procedures involving mice were approved by the institutional ethics committee of the Université du Québec à Montréal (CIPA protocols #650) in accordance with the biomedical research guidelines of the Canadian Council of Animal Care. The *Fam172a*$^{Tp/+}$ (FVB/N background; also known as *Toupee*$^{Tg/+}$) mouse line was as previously described (4). Embryos were generated by natural mating and staged by considering noon of the day of vaginal plug detection as e0.5. Control and mutant embryos were obtained from the same litter and processed in parallel. Primers used for genotyping are listed in Table S1.

## DNA constructs and recombinant proteins

MYC-, MBP-, and HA-tagged versions of the 371aa-long isoform of murine FAM172A (WT and E229Q-mutated) were as previously described (4). Vectors containing human *AGO1* (pcDNAM-human

eIF2C1-WT-Myc; gift from Dr. K Ui-Tei) (50) and *AGO2* ORFs (p3XFLAG-myc-CMV-AGO2, gift from Dr. E Chan) (51) were obtained from Addgene (#50360 and #21538, respectively). For BiFC assays, full-length ORFs of *Fam172* and *AGO2* were subcloned in expression vectors bearing either the N-terminal (pCDNA3-$_{1-173}$Venus) or the C-terminal half (pCDNA3-$_{174-259}$Venus) of Venus, which were both generously provided by Dr. S Merabet (Institut de génomique fonctionnelle de Lyon) (52). NLS-mutated FAM172A (R227Q, R228Q, R241Q, and R242Q), phosphodead FAM172A (D219Q, D222Q), phosphomimic FAM172A (S215D, S216D, S217D, S218D, and T222D), FLAG-tagged versions of both AGO1 and AGO2 and NLS-containing $_{FLAG}$AGO2 were all generated using the Gibson Assembly method, as previously described (53, 54). Briefly, relevant PCR amplicons (see Table S1 for the primers' list) and digested plasmid (ratio 10:1) were added to the 1X Gibson Master Mix (5% PEG-8000, 100 mM Tris–HCl, pH 7.5, 10 mM MgCl2, 10 mM DTT, 200 $\mu$M each of the four dNTPs, and 1 mM NAD) and then combined to 0.01 U of T5 EXO, 0.06 U of Phusion polymerase and 10 U/$\mu$l Taq ligase for a 1-h-long incubation at 50°C. All DNA constructs were verified via Sanger sequencing.

## Recombinant proteins and in vitro kinase assay

Recombinant MBP, $_{MBP}$FAM172A, and $_{MBP}$FAM172A[E229Q] proteins were produced in BL21 bacteria via IPTG (0.3 mM) induction of pMAL-c5X constructs, purified using amylose affinity chromatography and eluted in column buffer, as previously described (4). Recombinant His-tagged versions of AGO1 and AGO2 proteins were purchased from Sino Biological (#11225-H07B #11079-H07B, respectively). Recombinant CK2 was purchased from New-England Biolabs (P6010S) and used to phosphorylate recombinant FAM172A (with MBP tag released by factor Xa-mediated cleavage) in a sodium phosphate buffer supplemented with 50 mM KCl, 10 mM MgCl2 and 200 $\mu$M ATP (pH 7.5) for 1 h at 30°C. Samples were then immediately processed for mass spectrometry analysis by the Proteomics Discovery Platform of the Institut de Recherches Cliniques de Montréal.

## Cell culture and transfection

Adherent murine cells were grown in EMEM (N2a, MEFs) or DMEM (NIH3T3, R1) medium supplemented with either 10% (N2a, MEFs and NIH3T3) or 20% (R1) FBS, and 1% penicillin/streptomycin (all products from Wisent). MEFs were generated from e10.5 embryo heads dissociated in EMEM containing 1.3 mg/ml Dispase II (#17105–041; Life Technologies), 0.4 mg/ml collagenase (#C2674; Sigma-Aldrich), and 0.1 mg/ml DNase I (#DN25; Sigma-Aldrich), for 30 min at 37°C with gentle agitation. The resulting cell suspension was filtered through a 70-$\mu$m cell strainer (Fisherbrand) and then plated on gelatin-coated coverslips for immunofluorescence or gelatin-coated plates for protein extraction. Human lymphoblastoid cell lines derived from a FAM172A[E228Q]-associated CHARGE family (4) were grown in suspension in RPMI medium supplemented with 10% FBS and 1% penicillin/streptomycin (all products from Wisent). All cell cultures were maintained at 37°C and 5% $CO_2$ in a humidified atmosphere. All transfections were performed using GeneJuice reagents (#70967; Millipore-Sigma) in accordance with the manufacturer's instructions (0.1 $\mu$g of

DNA per cm$^2$ of cultured cells, and 1 µg of DNA for 3 µl of GeneJuice reagent).

## Immunofluorescence analysis of cultured cells

Cells plated on coverslips were transfected with the appropriate plasmid for 48 h, fixed in 4% paraformaldehyde (diluted in PBS) for 15 min on ice, and washed three times in PBS. The cells were then incubated in a blocking solution (10% FBS and 0.1% TritonX-100 diluted in PBS) for 1 h at room temperature, followed by overnight incubation at 4°C with primary antibodies of interest diluted in the blocking solution (see Table S2 for antibodies list). After three more washes in the blocking solution, the cells were incubated with relevant secondary antibodies conjugated with AlexaFluor 488, 594 or 647 for 1 h at room temperature (see Table S2 for antibodies list), washed again three times, and finally counterstained with 5 µg/ml DAPI (40,6-diamidino-2-phenylindole) for 10 min. Images were acquired using a Nikon A1 laser scanning confocal microscope with Plan Apo λ 60 × 1.40 objective and analyzed using ImageJ. Mean fluorescence intensity in the nucleus and the cytoplasm was quantified using the *measure* function and these values were used to evaluate the relative abundance of proteins of interest in each cell compartment, which was expressed in log$_2$ scale (log$_2$[N:C] ratio).

## Cell fractionation

Cell fractionation of MEFs grown at high density (200,000 cells/cm$^2$) and exposed to 2 ng/ml LMB (#9676; Cell Signaling Technology) during 4 h was performed using the REAP method (55). Briefly, cells were collected by trypsinization, gently centrifuged, and washed twice with PBS. Cell pellets were subsequently resuspended in ice-cold 0.1% NP-40 diluted in PBS and triturated five times. One-third of this extract was kept as the "total fraction," whereas the remaining was centrifuged to pellet the nuclei. The supernatant was then kept as the "cytoplasmic fraction," whereas the pelleted nuclei were washed once with 0.05% NP-40 diluted in PBS, and directly resuspended in standard Laemmli loading buffer to constitute the "nuclear fraction." Preparation of both the "total fraction" and the "nuclear fraction" was finally completed by sonication (Qsonica, 30% amplitude, 2 × 30 s).

## co-IP in cells and in vitro

Co-IP assays in cells were performed using two confluent 100 mm plates, 48 h after transfection with relevant expression vectors. Whole-cell extracts were prepared in a defined lysis buffer with moderate ionic strength (50 mM Tris pH 8.0, 100 mM NaCl, 1 mM EDTA, 1% Triton TX-100, 1X Roche Complete protease inhibitors) during 1 h at 4°C, after which, the samples were centrifuged to remove insoluble material. To enhance co-IP of endogenous importins and FAM172A dimers, cells were crosslinked before lysis during 10 min with 0.75% formaldehyde, which was then quenched during 5 min using 0.125 M glycine, in accordance with previous studies that used this approach to capture presumably transient protein–protein interactions (56, 57, 58). In vitro co-IP

assays were performed using 300 ng of purified recombinant proteins diluted in a low-salt buffer (20 mM Tris pH 8.0, 25 mM NaCl, 1 mM EGTA, 1 mM EDTA, 1.5 mM MgCl2, 1 mM DTT, 1% Triton TX-100, 10% glycerol, 1X Roche Complete protease inhibitors) previously described to enhance FAM172A–AGO2 interaction (4). For both cellular and in vitro assays, 10% of protein mixture was kept aside as input material and the rest was incubated with a primary antibody of interest overnight at 4°C (see Table S2 for the antibodies list). Antibody-bound complexes were then isolated using protein G-coupled magnetic beads (Dynabeads protein G, Thermo Fisher Scientific) and analyzed either by Western blot (see below) or mass spectrometry (for assessing FAM172A phosphorylation) (4).

## Western blot

Protein samples were separated on SDS–PAGE mini-gels using the Mini-PROTEAN system (Bio-Rad Laboratories) and then transferred to PVDF membranes. The membranes were blocked at room temperature for 1 h in TBST (Tris-buffered saline-Tween 20) containing 5% nonfat milk, followed by overnight incubation at 4°C with primary antibody of interest (see Table S2 for the antibodies list). After three washes with TBST, the membranes were subsequently incubated with relevant HRP-conjugated secondary antibodies for 1 h at room temperature (see Table S2 for the antibodies list). Chemiluminescent detection was finally performed using Immobilon Western HRP substrate (MilliporeSigma) and captured using the Fusion FX imaging system (Vilber). When necessary, relative protein levels were quantified using the *gel analyzer* function in ImageJ.

## BiFC assays in cells

Efficiency of BiFC assays in six-well plates was assessed 24 h posttransfection by calculating the percentage of Venus (YFP)-positive cells and mean fluorescence intensity using a BD Accuri C6 flow cytometer (BD Biosciences). After reaching the desired number of detected events (30,000 or 100,000 cells), the remaining cells were used to verify abundance of N-Venus- and C-Venus-tagged proteins by anti-GFP Western blot (see below). Protein abundance was then used for normalization of fluorescence intensity, which also took into account auto-fluorescence in non-transfected cells as the background value to be subtracted for all samples. Subcellular distribution of BiFC signal was assessed 48 h posttransfection in unfixed cells plated in 35 mm µ-dishes (#81156; Ibidi), after a 5-min incubation with 5 µg/ml Hoechst 33342. For live imaging, cells were plated on eight-well chamber µ-slides (#80826; Ibidi), exposed to either 2 ng/ml LMB (#9676; Cell Signaling Technology), 5 µM ivermectin (#S1351; Selleckchem) or vehicle only (same volume of 100% ethanol or DMSO), and monitored for an additional 5 h under normal growing conditions, with image acquisition every 20 min. For CK2 inhibition, cells were treated with 20 µM 4,5,6,7-tetrabromobenzotriazole (#T0826; Sigma-Aldrich) 2 h before the start of monitoring. All images were acquired using a Nikon A1 confocal microscope and analyzed using ImageJ, as described above.

## RNA extraction and RT–qPCR

RNA from e10.5 MEFs was extracted using the RNeasy Plus Purification Kit (QIAGEN) in accordance with the manufacturer's protocol. cDNAs were then generated using 50 ng of total RNA and Superscript II reverse transcriptase (Thermo Fischer Scientific). qPCR experiments were finally carried out using the Ssofast EvaGreen Supermix and C1000 Touch thermal cycler (Bio-Rad Laboratories), with *Psmb2* gene used for normalization of absolute expression levels and constant exons of the gene of interest (*Cd44*) used for normalization of alternative splicing events (see Table S1 for the primers' list).

## Statistical analysis

Graphical data are represented as the mean ± SD, with the number of independent experiments (N) and/or independent biological replicates (n) specified in the figure and/or legend when relevant. Significant differences between samples were determined using the GraphPad Prism software version 6.0, with 0.05 as the statistical significance cutoff. Selected statistical tests are indicated in figure legends.

# Data Availability

All relevant data are within the manuscript and its supporting information files.

# Supplementary Information

# Acknowledgements

The authors thank the Cellular analyses and Imaging core (CERMO-FC, UQAM) for assistance with confocal imaging and flow cytometry; the Proteomics Discovery Platform (IRCM) for the mass spectrometry analyses; Dr. S Merabet and J Reboulet (Institut de génomique fonctionnelle de Lyon) for providing BiFC constructs and technical advice; Dr. JW Belmont and P Hernandez (Baylor College of Medicine) for providing the human lymphoblastoid cell lines; and Dr. K Borden and B Culjkovic-Kraljacic (Institut de recherche en immunologie et cancer) for the gift of importin antibodies. This work was supported by grants from the Canadian Institutes of Health Research to N Pilon (grant #PJT-152933 and OGB-183998). N Pilon was also supported by the Fonds de la recherche du Québec – Santé (FRQS Senior Research Scholar) and by the UQAM Research Chair on Rare Genetic Diseases, whereas E Leduc and C Bélanger were supported by doctoral scholarships from the Natural Sciences and Engineering Research Council (NSERC) and FRQS, respectively, and both S Sallis and F Azouz were supported by scholarships from the CERMO-FC.

## Author Contributions

S Sallis: data curation, formal analysis, investigation, methodology, and writing—original draft.

F-A Bérubé-Simard: investigation and methodology.
B Grondin: data curation, formal analysis, investigation, and methodology.
E Leduc: data curation and investigation.
F Azouz: data curation and investigation.
C Bélanger: data curation, investigation, and methodology.
N Pilon: conceptualization, supervision, funding acquisition, project administration, and writing—review and editing.

## Conflict of Interest Statement

The authors declare that they have no conflict of interest.

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
