## [Reviewer comments · Life Science Alliance]

Life Science Alliance

The CHARGE syndrome-associated protein FAM172A controls AGO2 nuclear import

Sephora Sallis, Félix-Antoine Bérubé-Simard, Benoit Grondin, Elizabeth Leduc, Fatiha Azouz, Catherine Bélanger, and Nicolas Pilon

DOI: <https://doi.org/10.26508/lsa.202302133>

Corresponding author(s): Nicolas Pilon, University of Quebec at Montreal

Review Timeline:

Submission Date:	2023-05-04
Editorial Decision:	2023-05-04
Revision Received:	2023-05-08
Editorial Decision:	2023-05-09
Revision Received:	2023-05-10
Accepted:	2023-05-11

Transaction Report:

Please note that the manuscript was previously reviewed at another journal and the reports were taken into account in the decision-making process at Life Science Alliance.

Reviewer #1 Review

Comments to the Authors (Required):

The authors have addressed all my concerns with explanations and new experiments. I am happy to recommend acceptance.

Reviewer #2 Review

Comments to the Authors (Required):

In the revised version of their manuscript, Sallis, Pilon and coworkers address the points that were raised by this reviewer - without, however, diving much deeper into the molecular mechanisms of AGO2-import. Clearly, FAM172A plays a role, but the authors point out that FAM172A is probably not the only mediator of AGO2 nuclear import and that FAM172A-mediated nuclear import of AGO2 may also occur via other pathways than the canonical importin alpha/beta pathway. The import complex remains ill-defined, as "FAM172A can co-329 immunoprecipitate importin- α 1/ β 1 even when using our NLS-mutated version (Fig.S7C-D)". This may or may not be related to the buffer conditions used for the IP-experiments (the authors do address the low-salt-issue that came up after the initial submission. Still, I think the results of such experiments (using 25 or 100 mM NaCl and/or using crosslinkers) have to be interpreted with caution). Also, the ivermectin effect could be indirect, as it takes a rather long time to see nuclear import in the presence of LMB (2-3 h; Fig. 3g).

As stated before, the findings are well described and the authors convincingly show that "The CHARGE syndrome-associated protein FAM172A controls AGO2 nuclear import". The study still lacks, however, novel and significant mechanistic insight into the underlying transport pathway(s).

Reviewer #3 Review

Comments to the Authors (Required):

The response by the authors falls short of reassuring me, indeed most of my points have no satisfactory response. This is a field

that has been the subject of conflicting and unreliable previous studies. On re-reading the paper, I believe the fundamental problem is that much of the data is either indirect or descriptive. The authors over-interpret the data - injecting too much certainty into their conclusions. In both the Response and the manuscript, the language often lacks scholarly restraint and goes beyond what the data justify. Many of the experiments are also complex and have ambiguous interpretations. I acknowledge that these are difficult experiments and that authors should be encouraged to publish the data they have, not the data we might wish them to have if they were to expend months of effort responding to reviewers. A manuscript that better describes the limitations of the experimental approaches and has more modest and thoughtful conclusions might be a better addition to the confused literature on this topic.

Reviewer #1 Review

Comments to the Authors (Required):

CHARGE syndrome is a neural crest disorder mainly caused by mutation of the chromatin remodeler-coding gene CHD7. Alternative causes include mutation of other chromatin and/or splicing factors. One of these additional players is the poorly characterized FAM172A, which the authors previously found in a complex with CHD7 and the small RNA-binding protein AGO2 (Argonaute-2) at the chromatin-spliceosome interface.

AGO2 is a small RNA-binding protein best known as a major component of the RNA induced silencing complex (RISC), which orchestrates post-transcriptional gene silencing in the cytoplasm. However, there are increasing lines of evidence supporting non-canonical roles of AGO2 in the nucleus. Most of the work linking AGO proteins and their nuclear functions has been performed in plants and yeast but little is known about the potential nuclear AGO protein functions in mammals. In agreement with these canonical and non-canonical functions, AGO2 has been reported to shuttle between the cytosol and the nucleus of mammalian cells. However, the molecular mechanism behind AGO2 nuclear import remains unknown.

In this work, the authors showed by combining biochemistry and microscopy approaches that the NLS-containing FAM172A regulates AGO2 nuclear import via a direct protein-protein interaction, which is influenced by the status of CK2-mediated phosphorylation of FAM172A. In addition, perturbation of this process is functionally relevant, contributing to some of the cellular and molecular defects previously identified in the Fam172aTp/Tp 102 mouse model of CHARGE syndrome. Overall, the experiments are well designed and the conclusions shed light about the mechanism that imports AGO2 into the nucleus. Moreover, this study strengthens the notion that non-canonical nuclear functions of AGO2 and associated regulatory mechanisms are clinically relevant in human disease. However, there are critical points that require further clarification to support the proposed molecular mechanism:

1) In figure 2, the authors showed by coimmunoprecipitation experiments that the PAZ domain-containing N-terminal half of AGO2 as being essential for mediating the interaction with FAM172A. However, when the authors evaluated the functional relevance of this direct interaction for AGO2 nuclear localization they did not test the localization of this construct either by biochemical fractionation or microscopy analysis. The prediction from the proposed model is that the deletion of the PAZ domain should impair the import of AGO2 into the nucleus.

2) In figure 4, the authors concluded by mutational studies of FAM172A that CK2-mediated phosphorylation of FAM172A is important for FAM172A-dependent nuclear import of AGO2. If the proposed working model is correct CK2 knockdown should reduce AGO2 nuclear import

3) In figure 5, the authors showed the functional relevance of FAM172A-regulated nuclear import of AGO2 via rescue experiments in Fam172aTp/Tp MEFS, by combining cell proliferation assays and splicing defects. To this end, the authors tested FAM172A and its mutant versions or AGO2. The authors concluded that overexpression of WT AGO2 (which increases overall levels in both cytoplasm and nucleus) partially rescued while another version engineered to contain a strong NLS (3 copies of SV40 monopartite NLS) fully rescued the proliferation rate of Fam172aTp/Tp MEFS. However, to reinforce the working model the authors should test the PAZ domain deletion evaluated in figure 2 to demonstrate that AGO2 nuclear import reduction is unable to rescue cell proliferation and splicing defects.

Minor point:

In legend of figure 1, it is not clear what N and n stand for.

Reviewer #2 Review

Comments to the Authors (Required):

In their manuscript „The CHARGE syndrome-associated protein FAM172A controls AGO2 nuclear import" Sallis, Pilon and coworkers investigate nuclear import of the RISC-component Argonaute-2 (AGO2), which, despite its very prominent function(s) has not been elucidated so far. Of note, AGO-2 does not contain a (recognizable) NLS or NES. The authors make the point that the protein FAM172A functions as a mediator of nuclear import, most likely allowing a "piggyback" mechanism for import of

AGO2. Furthermore, the authors investigate the effects of phosphorylation of FAM172A on nuclear import of the protein itself and on that of AGO2.

The findings are well described and technically of high quality. The major concern here is the nature of the postulated complex of FAM172A and AGO2. As the authors mention, BiFC can lead to stabilization of very transient/low-affinity complexes (lane 183). Likewise, the biochemical characterization of the complex (Fig. 1) is not very convincing. The authors use low-salt buffers for all their binding/IP-experiments. Are those interactions stable in buffers containing physiological salt concentrations? Would AGO1, which seems to be unaffected by FAM172A, interact in low-salt buffers? Also, the mutant E229Q seems to interact with AGO2 to some extent (Fig. 2A). The authors should characterize the postulated complex of FAM172A and AGO2 in more detail. This could also involve bona fide nuclear import complexes containing, for example, importin alpha (which was already identified as a potential binding partner of FAM172A (lane 276) and importin beta, leading to a real insight into the molecular mechanisms of AGO2-import. Without this, we are left with very interesting observations, pointing to a role of FAM172A in the subcellular localization of AGO2.

Minor points:

Fig. 1D: The method used for fractionation should be briefly described.

Fig. 1I: The log₂-ratio after rescue with Myc-F172A is around "0", whereas it is around 1.5 in WT cells in Fig. 1B. The authors should comment on this partial rescue. It seems that the "predominant nuclear localization of AGO2" is not "efficiently re-established" (as stated in the manuscript (lane 137-139)).

Fig. 2A: Better describe the Co-IP experiment. Anti-MBP was used here? Why is immunoprecipitation used in the first place? Recombinant proteins are used, which could be immobilized on beads. What is meant by "lysate" (lane 527 in methods-section)? Also, Coomassie gels would be more convincing.

Figs. 1I, 2E and 3H are partially redundant.

Reviewer #3 Review

Comments to the Authors (Required):

Argonaute is an important protein in the RNAi machinery. While Ago is often thought to function in the cytoplasm, it may also act in cell nuclei. Exactly how AGO shuttles between cytoplasm and nuclei is unknown in spite of several papers claiming to offer insights. The strength of this manuscript is that it addresses an understudied research problem. The weaknesses is that the data are not sufficiently persuasive.

- 1) This is not a "CHARGE" syndrome paper. The introduction needs to be rewritten to focus on the main point of the paper.
- 2) In Figure 1, the microscopy results are qualitative. They do contribute to the paper, but are offset by the small change in Ago localization shown in the western (1C). Clearly, AGO can still enter the nucleus. The quantitative measures of Ago (total Ago and Ago in cytoplasm versus nucleus) are not well explained.
- 3) No western data is provided for the second part of Figure 1.
- 4) The question arises, if Ago localization is changing (not clear enough to this reviewer) is it a direct or indirect affect of mutating FAM172A.
- 5) The experiments in Figure 2 use recombinant proteins, which might overload the system and yield misleading results. Also, at best, the experiments suggest that FAM172A and Ago might interact inside cells.
- 6) I'm troubled that Figure 4 does not have western analysis of AGO localization
- 7) The premise for the Figure 5 is not clear. The differences are small. use of Cd44 in these cells for such a pivotal experiment is problematic. Cell proliferation may have little to do with any Ago function. This figure is not really needed anyway, the point of the paper is Ago import, not its function once it is in the nucleus.

In summary, the lack of clear, decisive western analysis of protein localization reduces my enthusiasm for this paper because that deficiency prevents it from rising about previous papers that often conflicting results. Even if the correlation between FAM expression and Ago important was stronger, I would remain concerned about whether the effect was direct or indirect.

May 4, 2023

Re: Life Science Alliance manuscript #LSA-2023-02133-T

Prof. Nicolas Pilon
University of Quebec at Montreal
141 President-Kennedy avenue
Montreal H2X3Y7
Canada

Dear Dr. Pilon,

Thank you for submitting your manuscript entitled "The CHARGE syndrome-associated protein FAM172A controls AGO2 nuclear import" to Life Science Alliance. We invite you to submit a revised manuscript to acknowledge the limitations of the approach and temper the conclusions made, along the lines outlined by Reviewers 2 and 3.

Thank you for this interesting contribution to Life Science Alliance. We are looking forward to receiving your revised manuscript.

Sincerely,

B. MANUSCRIPT ORGANIZATION AND FORMATTING:

May 9, 2023

RE: Life Science Alliance Manuscript #LSA-2023-02133-TR

Prof. Nicolas Pilon
University of Quebec at Montreal
141 President-Kennedy avenue
Montreal H2X3Y7
Canada

Dear Dr. Pilon,

Thank you for submitting your revised manuscript entitled "The CHARGE syndrome-associated protein FAM172A controls AGO2 nuclear import". We would be happy to publish your paper in Life Science Alliance pending final revisions necessary to meet our formatting guidelines.

- please consult our manuscript preparation guidelines <https://www.life-science-alliance.org/manuscript-prep> and make sure your manuscript sections are in the correct order
- please use the [10 author names, et al.] format in your references (i.e. limit the author names to the first 10)

A. FINAL FILES:

B. MANUSCRIPT ORGANIZATION AND FORMATTING:

****It is Life Science Alliance policy that if requested, original data images must be made available to the editors. Failure to provide**

original images upon request will result in unavoidable delays in publication. Please ensure that you have access to all original data images prior to final submission.**

The license to publish form must be signed before your manuscript can be sent to production. A link to the electronic license to publish form will be sent to the corresponding author only. Please take a moment to check your funder requirements.

Sincerely,

May 11, 2023

RE: Life Science Alliance Manuscript #LSA-2023-02133-TRR

Prof. Nicolas Pilon
University of Quebec at Montreal
141 President-Kennedy avenue
Montreal H2X3Y7
Canada

Dear Dr. Pilon,

Thank you for submitting your Research Article entitled "The CHARGE syndrome-associated protein FAM172A controls AGO2 nuclear import". It is a pleasure to let you know that your manuscript is now accepted for publication in Life Science Alliance. Congratulations on this interesting work.

DISTRIBUTION OF MATERIALS:

Again, congratulations on a very nice paper. I hope you found the review process to be constructive and are pleased with how the manuscript was handled editorially. We look forward to future exciting submissions from your lab.

Sincerely,
